# Metabolic Dysfunction-Associated Steatotic Liver Disease as a Risk Factor for Chronic Kidney Disease: A Narrative Review

**DOI:** 10.3390/biomedicines13092162

**Published:** 2025-09-04

**Authors:** Marcelo do Rego Maciel Souto Maior, Nathália de Lacerda Interaminense Ribeiro, Hannah Vicentini Vitoriano Silva, Edmundo Pessoa Lopes, Emilia Chagas Costa

**Affiliations:** 1Postgraduate Program in Nutrition, Center for Health Sciences, Federal University of Pernambuco, Recife CEP 50670-901, Brazil; emilia.costa@ufpe.br; 2Department of Pathology, Federal University of Pernambuco, Recife CEP 50670-901, Brazil; nathalia.lacerda@upe.br (N.d.L.I.R.); hannahvicentini@gmail.com (H.V.V.S.); 3Department of Internal Medicine, Center for Medical Sciences, Federal University of Pernambuco, Recife CEP 50670-901, Brazil; epalopes@uol.com.br; 4Physical Education and Sports Science Unit, Vitoria Academic Center, Federal University of Pernambuco, Vitória de Santo Antão CEP 55608-680, Brazil

**Keywords:** metabolic dysfunction-associated liver disease, non-alcoholic fatty liver disease, chronic kidney disease, metabolic syndrome, obesity, diabetes mellitus

## Abstract

Metabolic dysfunction-associated steatotic liver disease (MASLD)—previously known as non-alcoholic fatty liver disease (NAFLD)—is currently the most common chronic liver disease globally. Observational studies have reported that MASLD is independently associated with extrahepatic disorders, such as chronic kidney disease (CKD). Severe forms of MASLD (i.e., steatohepatitis and liver fibrosis) are even more strongly associated with the risk of incident kidney dysfunction. Hypothetically, MASLD could directly promote CKD through liver-derived endocrine and metabolic mediators, hemodynamic alterations, immune-mediated mechanisms, and oxidative or cellular stress. However, proving that MASLD directly causes CKD is difficult due to the multiple shared cardiometabolic and systemic risk factors, such as obesity, hypertension, and type 2 diabetes mellitus, which serve as confounding variables. Moreover, studies on the association between MASLD and CKD have differed in their designs, sampling methods, disease definitions, and inclusion criteria, precluding more robust evidence supporting a causal relationship. Furthermore, few studies have explored specific issues, such as the new nomenclature for steatotic liver disease, the relationship between these diseases in pediatric populations, the impact of MASLD plus alcohol intake (MetALD) on CKD, and therapeutic options targeting MASLD and CKD simultaneously. Answers to these issues are essential, as the appropriate management of patients with MASLD may prevent or ameliorate kidney dysfunction. The aims of the present study are to describe shared risk factors between MASLD and CKD, the possible direct pathogenic effect of MASLD on kidney structure and function, and gaps in the current literature, to indicate future research directions.

## 1. Introduction

The term metabolic dysfunction-associated steatotic liver disease (MASLD) was recently proposed by a multisociety expert consensus as part of a new nomenclature system to replace non-alcoholic fatty liver disease (NAFLD) and metabolic dysfunction-associated fatty liver disease (MAFLD) [1]. Unlike NAFLD, which is defined by the exclusion of alcohol and other causes, MASLD applies inclusive criteria focused on pathophysiologically related metabolic dysfunction. Moreover, MASLD replaces the stigmatizing term “fatty” present in NAFLD and MAFLD with the more technical word “steatotic”. Thus, MASLD terminology has gained acceptance in both research and clinical settings [1,2].

Due to its rapidly increasing incidence, MASLD has become the most common chronic liver disease throughout the world, the prevalence of which is approximately 38% of the adult population [2]. Most cases of MASLD consist of isolated hepatic steatosis, but 12% to 40% of these may evolve to metabolic dysfunction-associated steatohepatitis (MASH), which presents with inflammation and ballooning degeneration of hepatocytes. Of these, 35% develop early hepatic fibrosis, usually beginning in a perisinusoidal pattern. In 15% of such cases, progression to advanced fibrosis or cirrhosis occurs (Figure 1) [3]. As a result, MASLD is now the second leading cause of liver transplantation in the United States and Europe [4].

Some extrahepatic disorders, such as cardiovascular disease, extrahepatic cancers, and chronic kidney disease (CKD), have also been shown to be associated with MASLD [5]. Indeed, the prevalence of CKD among the general population is around 10% but reaches 20–50% among individuals with MASLD, aggravating the long-term economic burden as well as increasing morbidity and mortality rates [6,7,8].

Chronic kidney disease (CKD) is defined by persistent abnormalities in kidney function or structure for at least three months, typically with progressive decline. Diagnosis is based on a sustained glomerular filtration rate (GFR) below 60 mL/min/1.73 m^2^ or evidence of kidney damage, such as albuminuria, abnormal urinary sediment, electrolyte or tubular disorders, structural abnormalities seen on imaging or histology, or a history of kidney transplantation [9].

Key histological features shared across all forms of CKD include interstitial fibrosis, tubular atrophy, global glomerulosclerosis, and arterial intimal thickening (Figure 2A) [10]. In contrast, certain findings may point to a specific underlying cause, such as diabetic nodular glomerulosclerosis (Figure 2B) [11].

MASLD and CKD share cardiometabolic risk factors, such as obesity, dyslipidemia, type 2 diabetes mellitus (T2DM), and hypertension [12,13]. However, MASLD has been shown to be independently associated with CKD even when adjusted for cardiometabolic covariates [8,14,15,16]. Hypothetically, MASLD could cause CKD through liver-induced specific dysregulations [17]. However, proving this independent association is difficult, as other systemic disorders of a genetic, dietary, infectious, or immunological nature have also been shown to predispose individuals to both MASLD and CKD. These disorders may serve as additional confounding variables that have not been uniformly included in studies [17,18,19,20,21,22,23,24].

This pathogenic complexity, along with nomenclature changes from NAFLD to MAFLD and then MASLD, has given rise to considerable diversity in study designs. Furthermore, some specific research issues have not yet been widely investigated, which is the case for pediatric populations, the impact of MASLD combined with alcohol intake (MetALD) on kidney dysfunction, and outcomes in patients treated for both MASLD and CKD [25,26,27].

Filling these gaps in knowledge and clarifying the interplay between MASLD and CKD would allow clinicians to manage patients accordingly. For instance, risk stratification and the prevention and treatment of MASLD may be warranted if this condition proves to be associated with an increased risk of CKD [8,28,29].

The aim of the present study was to conduct a literature review on this relationship. After a brief summarization of the new steatotic liver disease nomenclature and its implications in research and clinical practice, we discuss pathogenic links between MASLD and CKD, including mechanisms through which MASLD may directly cause CKD. Finally, we address some gaps in the literature and provide directions for future research.

## 2. Bibliographic Search Strategy and Selection Criteria

The authors searched the PubMed database up to June 2025, including only human studies published in English, with no restrictions on age range. We used the following terms: “metabolic dysfunction-associated steatotic liver disease” OR “MASLD” OR “ steatotic liver disease” OR “SLD” OR “alcoholic liver disease” or ALD” OR “MASLD and increased alcohol intake” OR “MetALD” OR “metabolic dysfunction-associated fatty liver disease” OR “MAFLD” OR “non-alcoholic fatty liver disease” OR “NAFLD” OR “non-alcoholic steatohepatitis” OR “NASH” OR “metabolic dysfunction-associated steatohepatitis” OR “MASH” AND “chronic kidney disease” OR “CKD”.

We performed additional queries using more specific terms: “epidemiology”, “pathogenesis”, “pediatric”, “children”, “management”, “obesity”, “diabetes”, “arterial hypertension”, and “metabolic syndrome”.

Non-peer-reviewed articles, theses, and preprints were excluded.

The final selection was based on the relevance and novelty of the articles. 

## 3. New Steatotic Liver Disease Nomenclature

Until 2020, fatty liver disease was divided into two mutually exclusive groups: NAFLD and alcoholic liver disease (ALD) [1,30]. The diagnosis of NAFLD was based on the exclusion of excessive alcohol intake and other competing causes of steatosis, such as viral hepatitis or drugs [31,32]. Despite its time-established use, the term NAFLD was criticized due to its focus on exclusionary criteria rather than emphasizing its relationship with metabolic syndrome [33].

In 2020, Eslam et al. suggested the term metabolic dysfunction-associated fatty liver disease (MAFLD) to replace NAFLD [34]. The diagnosis of MAFLD is entirely inclusive, requiring steatosis plus one major metabolic criterion (T2DM or overweight/obesity) or—in lean, non-diabetic individuals—at least two cardiometabolic risk abnormalities, irrespective of alcohol intake or other competing causes of steatosis [34]. This has led to criticisms, as metabolic and non-metabolic etiologies can coexist in the same patient, leading to a lack of specificity and diagnostic inaccuracies [33,35,36]. Moreover, the word “fatty” persisted in the MAFLD terminology, which is considered stigmatizing [1,30].

In 2023, steatotic liver disease (SLD) was recommended as a broad term to comprise the causes of liver steatosis [1]. SLD is divided into five subgroups: (1) MASLD (which includes major metabolic criteria but excludes alcohol consumption greater than 20 g/day for women and 30 g/day for men); (2) MetALD (consisting of MASLD plus alcohol intake of 20–50 g/day for women and 30–60 g/day for men); (3) ALD (average alcohol intake >50 g/day for women and >60 g/day for men); (4) specific etiology SLD (including monogenic diseases, drug-induced liver injury, and viral hepatitis); and (5) cryptogenic SLD [1]. This terminology improves specificity and substitutes the word “fatty” for the more technical word “steatotic” [1].

As the diagnostic criteria for these three nomenclature systems are slightly different, the impact of the new MASLD criteria on research related to CKD has not yet been widely tested [2,37,38]. However, NAFLD, MAFLD, and MASLD are all related to metabolic syndrome, and diagnostic agreement among these classifications is greater than 90%, according to some studies [33,38,39]. Therefore, the terms will be equivalent in this review and used in accordance with the respective references.

## 4. Shared Risk Factors Between MASLD and CKD

Metabolic syndrome-related conditions—such as obesity, insulin resistance, T2DM, hypertension, and dyslipidemia—are known to promote MASLD and CKD [13]. Other factors—including genetic polymorphisms, nutritional aspects, gut dysbiosis, aging, platelet activation, and sarcopenia—have more recently proved to be associated with MASLD and CKD, often with a bidirectional cause-and-effect relationship [7,40,41,42,43,44] (Figure 3 and Table 1).

### 4.1. Metabolic Syndrome

The prevalence of MASLD and CKD among individuals with obesity is 75% and 33%, respectively, which is much higher than the prevalence of these diseases in the general population (38% and 10%, respectively) [65,66]. Furthermore, individuals with MASLD and obesity have higher proportions of MASH (34%) and significant fibrosis (20%) [66,67]. Ectopic fat accumulation in the liver and kidneys may cause organ damage by direct compression or production of inflammatory cytokines, oxidative stress, procoagulant and fibrogenic factors, increased levels of leptin, and decreased levels of adiponectin [21,45,46,47]. Central obesity has been associated with the dysfunction of peroxisome proliferator-activated receptor-gamma (PPAR-γ), which is a regulator of fat storage and adipogenesis [46]. All these factors ultimately lead to increased insulin resistance, endothelial activation, thrombosis, fibrosis, hemodynamic disorders, and inflammation, aggravating both MASLD and CKD [25,38].

Insulin resistance and T2DM are also strongly linked to MASLD and CKD. According to a meta-analysis, the prevalence of NAFLD among individuals with T2DM is around 68%, which is almost twice that of the general population [68]. The prevalence of steatohepatitis (66%) and advanced fibrosis (15%) in this group is also higher than in non-diabetics, reflecting the further detrimental impact of insulin resistance on the progression of this liver disease [48,49].

T2DM is the most common etiology of CKD [69]. Insulin resistance stimulates fat accumulation in the kidney parenchyma and contributes to renal artery atherosclerosis, leading to macrovascular complications [11]. Hyperglycemia, in turn, generates advanced glycation end-products, promoting microvascular disease and glomerular injury, including nodular glomerulosclerosis, leading to proteinuria and the deterioration of the estimated glomerular filtration rate (eGFR) [50].

Atherosclerotic dyslipidemias and hypertension lead to the activation of the renin–angiotensin–aldosterone system [47,51]. Angiotensin II exacerbates insulin resistance and induces the progression of MAFLD and CKD through oxidative stress, parenchymal inflammation, and fibrosis [70]. Once installed, MASLD and CKD may worsen insulin resistance, dyslipidemia, and hypertension through inflammation, oxidative stress, uremia, metabolic acidosis, sedentarism, and further activation of the renin–angiotensin–aldosterone system [7].

### 4.2. Genetic Polymorphisms

Some genetic polymorphisms have been linked to NAFLD and CKD, but the most relevant is the rs738409 C > G mutation in the patatin-like phospholipase domain-containing 3 gene (PNPLA3), which is present in liver and kidney tissues [24,43]. This gene variant causes lipid accumulation and organ inflammation and is associated with a higher risk of NAFLD/steatohepatitis and liver fibrosis, as well as renal glomerular and tubular lesions, independently of metabolic factors [52].

Beyond *PNPLA3*, several genetic variants have been implicated in the shared pathophysiology of metabolic dysfunction-associated steatotic liver disease (MASLD) and chronic kidney disease (CKD). Notably, the rs58542926 variant of the Transmembrane 6 superfamily member 2 (TM6SF2) gene, and the variant rs641738 of the Membrane-bound O-acyltransferase domain-containing 7 (MBOAT7) gene, are associated with increased hepatic steatosis and fibrosis, with emerging evidence linking them to renal impairment, possibly through systemic inflammation and lipid dysregulation. The Glucokinase regulator (GCKR) (rs1260326) variant, which enhances hepatic lipogenesis via altered glucose metabolism, contributes to hepatic fat accumulation and may indirectly influence renal function through insulin resistance [53].

### 4.3. Nutritional Aspects and Gut Dysbiosis 

Hypercaloric diets can result in lipid accumulation in the liver and kidneys [20,44]. Fructose, which is present in sweeteners, is particularly harmful, as its metabolism generates uric acid, promoting insulin resistance, dyslipidemia, and nephrotoxicity, contributing to further fat deposition in the liver [46].

Gut dysbiosis occurs when the microbiota undergoes a reduction in its normal diversity, which is usually associated with Western diets and metabolic syndrome [71]. In gut dysbiosis, the predominance of harmful bacteria, along with a reduction in beneficial bacterial populations, results in the production of nephrotoxins and hepatotoxins [20,72].

Dietary choline is converted to trimethylamine-N-oxide (TMAO) by harmful gut microbiota. TMAO promotes atherosclerosis, the activation of inflammatory pathways, and oxidative stress, potentially stimulating renal interstitial fibrosis [20,21]. TMAO also inhibits the farnesoid X receptor, which is a hepatic bile acid receptor that regulates bile acid homeostasis and balances glucose and lipid metabolism in multiple tissues [54,72].

Bacterial toxins may also damage tight junctions in the gut epithelia, increasing mucosal permeability. Thus, pro-inflammatory cytokines and lipopolysaccharides (endotoxins) enter portal and systemic circulation. After reaching the liver, lipopolysaccharides activate Toll-like receptors (TLR) on hepatocytes, Kupffer cells, and stellate cells, leading to hepatic fibrosis [55]. In the kidneys, TLR activation by endotoxins leads to fibrosis and inflammation. Moreover, systemic urea levels rise with the impairment of renal function, further increasing intestinal permeability, thus engendering a vicious cycle [56,69].

### 4.4. Aging

Low antioxidant capacity and mitochondrial function are common in older people [41,42]. Together, these conditions promote inflammation, fibrosis, and fat accumulation in individuals with NAFLD as well as renal interstitial fibrosis and atherosclerosis [10,41,42].

Furthermore, aging is associated with sarcopenia, decreased levels of adiponectin (an insulin sensitizer), and increased liver secretion of fetuin-A, which is a protein associated with insulin resistance, inflammation, adipogenesis, and atherogenesis [57].

Other age-related phenomena include damage to sinusoidal endothelial cells in the liver, increased serum levels of uric acid, and decreased urinary levels of klotho protein, which is a marker of vascular calcification and the progression of CKD [38].

### 4.5. Platelet Activation

Platelet activation also has a bidirectional relationship with NAFLD and CKD and is intertwined with cardiometabolic disorders. For instance, oxidative stress, very low-density lipoprotein cholesterol, and intestinal dysbiosis all contribute to excessive platelet activation [13,58]. Activated platelets secrete pro-inflammatory cytokines (IL-6 and TNF-alpha) and growth factors, such as endothelial growth factor, platelet-derived growth factor, insulin-like growth factor 1 (IGF-1), transforming growth factor (TGF)-beta, and fibroblast growth factor (FGF) [58,59]. These molecules promote oxidative stress, hypercoagulation, endothelial activation, and inflammation, ultimately leading to atherosclerosis and parenchymal fibrosis in the kidneys and liver [60]. In the opposite direction, MASLD and CKD promote platelet activation by inducing inflammation, endothelial activation, oxidative stress, and in CKD, the accumulation of uremic toxins [40].

### 4.6. Sarcopenia

Another example of a bidirectional detrimental effect is found between MASLD/CKD and sarcopenia [40]. Sarcopenic muscles release myostatin, which is a member of the TGF-beta superfamily that can promote liver inflammation and the activation of hepatic stellate cells, culminating in liver fibrosis [61]. On the other hand, steatotic livers do not deal with energy substrates properly, often resulting in the catabolism of amino acids that would otherwise be used in muscle anabolism, thus aggravating sarcopenia [40,62]. Likewise, pro-inflammatory cytokines released by the liver accelerate muscle catabolism [62]. In cases of cirrhosis induced by NAFLD, hyperammonemia may aggravate the condition by inducing autophagy in myocytes [63].

Systemic inflammation and insulin resistance induced by sarcopenia worsen renal function [19]. In the opposite direction, CKD induces sarcopenia through low-grade systemic inflammation, protein loss due to dialysis, accelerated protein catabolism, and reduced anabolism, along with metabolic acidosis and sedentarism [64].

## 5. MASLD as a Possible Direct Cause of CKD

Hypothetically, MASLD may alter kidney structure and function through the pathophysiological mechanisms represented in Figure 4 [7,8,17]. Consistently, several studies have shown that the presence and severity of MASLD are associated with CKD, independently of many of the shared risk factors mentioned above [14,15,73,74,75,76,77,78,79,80]. Table 2 summarizes the results of the meta-analyses and longitudinal studies supporting this evidence.

As illustrated in Table 2, even when adjusted for cardiovascular comorbidities, MASLD was associated with increased incidence of CKD, and the hazard ratios were even greater when MASH or fibrosis were present [14,15].

These data from literature reinforce that MASLD may directly affect the renal parenchyma through intrinsic liver-driven mechanisms, as shown in Figure 4 and described below [7,8,17].

### 5.1. Endocrine and Metabolic Mediators

NAFLD not only develops as a result of insulin resistance but also actively contributes to its progression, establishing a vicious cycle [81]. Hepatic fat accumulation leads to the production of lipotoxic intermediates, pro-inflammatory cytokines, and hepatokines, which impair insulin signaling and disrupt insulin receptor pathways. Additionally, mitochondrial dysfunction and oxidative stress, along with endoplasmic reticulum stress, exacerbate these effects [47,58,82]. Notably, insulin resistance induced by NAFLD has been shown to promote macrovascular and microvascular complications that lead to proteinuria and CKD, even before the onset of diabetes [83,84]. If overt diabetes emerges during the course of NAFLD, the decline in renal function may be even faster; indeed, diabetes is the main etiology of CKD [11,50].

Along with insulin resistance, liver-generated atherogenic dyslipidemia, characterized by elevated levels of triglyceride-rich lipoproteins, small dense LDL particles, and reduced HDL cholesterol, contributes to macrovascular renal complications (atherosclerotic plaques), as well as systemic inflammation, endothelial dysfunction, and oxidative stress, all of which promote glomerular injury and tubulointerstitial fibrosis [47,58,85].

MASLD also alters liver secretion of hepatokines (hormone-like proteins that regulate systemic metabolic processes), aggravating renal injury. Hepatokines such as fetuin-A promote renal insulin resistance and inflammation, while proteins like angiopoietin-like protein 8 (ANGPTL8) and selenoprotein *p* are implicated in lipid imbalance and oxidative stress, exacerbating nephron damage [40,57,86].

Lastly, steatotic livers produce uremic toxins similar to those associated with gut dysbiosis, notably indoxyl sulfate, p-cresyl sulfate, and trimethylamine N-oxide (TMAO). Once in the bloodstream, these toxins accumulate in renal tissue, where they induce oxidative stress, and pro-inflammatory and pro-fibrotic signaling pathways, including those mediated by transforming growth factor-beta (TGF-β) [21].

### 5.2. Hemodynamic Alterations

MASLD can lead to subclinical portal hypertension even in the absence of cirrhosis, in a kind of hepatorenal reflex [87]. Indeed, increased intrahepatic vascular resistance may occur in pre-fibrotic stages of NAFLD as a consequence of fat accumulation and necro-inflammatory changes [88]. These changes may activate the renin–angiotensin–aldosterone system (RAAS) and the sympathetic nervous system, causing disturbances in renal perfusion, with consequent glomerular hyperfiltration and endothelial injury [70]. If hepatic fibrosis ensues, portal hypertension worsens, which is in line with evidence showing that NAFLD with fibrosis is more associated with the risk of developing CKD than steatosis alone [89]. Nevertheless, the hepatorenal reflex hypothesis, although interesting, requires further exploration in experimental and clinical research [40,90].

### 5.3. Immune-Mediated Mechanisms

MASLD, particularly when it progresses to MASH, leads to the intrahepatic production and systemic release of pro-inflammatory mediators such as tumor necrosis factor-alpha (TNF-α), interleukin-6 (IL-6), interleukin-1β (IL-1β), and C-reactive protein (CRP) [59,82]. These molecules circulate and act at distant sites, in a paracrine-like effect of liver inflammation, known as the “inflammatory spillover” hypothesis, which could explain how MASLD drives systemic immune activation [40,88,91]. In the kidney, these mediators may promote endothelial dysfunction, immune cell infiltration, mesangial expansion, and interstitial fibrosis [21,46].

Since inflammation and the coagulation cascade are closely linked, pro-coagulant factors—such as plasminogen activator inhibitor-1 (PAI-1), tissue factor, and thrombin—are usually activated as a consequence of the MASLD inflammatory environment [59,91]. PAI-1 inhibits fibrinolysis and contributes to renal interstitial fibrosis, while thrombin and tissue factor induce endothelial damage and microvascular thrombosis, leading to progressive obliteration of the renal microvasculature that supplies tubules and glomeruli, possibly contributing to nephron loss [18,90].

### 5.4. Oxidative and Cellular Stress Pathways

Steatosis and the necro-inflammatory alterations present in MASLD are significant sources of reactive oxygen species (ROS) due to mitochondrial overload and dysfunction. These ROS can be released into systemic circulation, damaging circulating proteins and lipoproteins and promoting lipid peroxidation [91,92,93]. In renal endothelial and tubular epithelial cells, ROS can lead to inflammation, apoptosis, and fibrosis [10,90].

Endoplasmic reticulum (ER) stress also occurs during MASLD progression, triggered by excessive free fatty acids and unfolded protein accumulation in hepatocytes. This results in systemic release of misfolded proteins and lipotoxic intermediates (e.g., diacylglycerols and ceramides) that can be harmful to renal tubular cells [7,17].

Importantly, ER stress and oxidative stress can both activate signaling pathways like the nuclear factor kappa-light-chain-enhancer of activated B cells (NF-kB), upregulating immune cell recruitment and stimulating the production of pro-inflammatory cytokines, hence forming a self-perpetuating loop with inflammation and further damaging the liver and kidneys [21,59,82].

## 6. Gaps and Limitations in Literature

Coupled with the complexity of the pathogenic mechanisms linking MASLD and CKD, gaps and limitations in the literature further hamper the elucidation of the relationship between these conditions [7,8,17].

### 6.1. Heterogeneity Among Studies

Designs, sampling methods, disease definitions, and inclusion criteria have varied among studies [14,38]. This is exemplified by the results of a recent meta-analysis investigating the association between MAFLD and CKD, in which high heterogeneity was found among both cross-sectional and longitudinal studies (I^2^ = 97.7% and I^2^ = 84.6%, respectively) [14]. This interstudy diversity precludes proper comparisons between individual study results.

One of the main causes of heterogeneity is the multitude of shared risk factors between MASLD and CKD, which act as confounding variables. Studies have considered traditional cardiometabolic covariates but have not adjusted for other risk factors in multivariate analyses, such as gut dysbiosis, unhealthy diets, old age, and sarcopenia [42,61,94,95].

Another source of divergence among studies is the variation in sample sizes and diagnostic definitions. Large population-based studies have used screening methods for the diagnosis of NAFLD/MAFLD, such as ultrasound or serum biomarkers [79,96,97,98,99,100]. Although widely available, ultrasound is not sufficiently sensitive to detect steatosis levels of less than 20% and is inaccurate in individuals with obesity [101]. Low specificity is found with serum biomarkers due to being based on variables such as platelet count, liver enzymes, obesity, and diabetes status, all of which can be altered in multiple conditions other than NAFLD [102]. Moreover, few studies have used liver histology due to the invasiveness of biopsies, despite histology being the gold standard for the diagnosis of NAFLD, persisting as the only tool capable of distinguishing steatohepatitis from steatosis alone [92]. Such studies have usually had a cross-sectional design, a small number of participants, and very specific inclusion criteria, impeding the generalization of the results [88,103,104,105,106].

Regarding such diagnostic issues, emerging multi-omics non-invasive biomarkers, with enough accuracy for MASLD diagnosis and detection of fibrosis, would be of great relevance [43,60].

Inclusion criteria for CKD have also been different among studies. Some studies have excluded albuminuria from the definition of CKD, whereas others have not [14,15]. Secondly, most studies define CKD as an eGFR < 60 mL/min/1.73 m^2^, which corresponds to CKD stages 3 to 5 [69,107]. Consequently, the relationship between NAFLD and early stages of CKD (1 and 2) is under-investigated [17]. Thirdly, eGFR is a proxy equation—based on age, gender, weight, and serum creatinine—to estimate the actual measured GFR (mGFR) [107]. Although mGFR is the gold standard for measuring renal function, it has not been used in research due to its cost and impracticality [10]. However, eGFR has a 30% rate of discordance with mGFR. For instance, eGFR underestimates mGFR in individuals with obesity and overestimates mGFR in older women and individuals with cirrhosis [107]. Lastly, as renal histology has not been used, we do not know if a specific kind of kidney injury is preferentially associated with MASLD [10,108].

With regards to ethnicity, most studies on the association between NAFLD and CKD have been conducted in Asia, the US, and Western Europe and have reported different results [109,110]. In contrast, few studies have been conducted in other regions, such as Latin America, where the prevalence of NAFLD reaches 44.4% [2].

To summarize, future studies should prioritize prospective cohort designs, with homogeneous diagnostic criteria for both MASLD and CKD, while also adjusting for as many confounding factors as possible, including gut dysbiosis and sarcopenia [8,57,71].

Lastly, although longitudinal studies have shown a higher incidence of CKD in individuals with MASLD, the bidirectional association between cardiometabolic disorders and these diseases raises the question of whether the opposite might also occur—i.e., CKD may be a predisposing factor for MASLD [25,46,111]. Indeed, CKD is known to promote dysglycemia, dyslipidemia, atherosclerosis, sarcopenia, and systemic inflammation—all of which are causes of liver steatosis [64,85,112]. In this respect, studies comparing the incidence of MASLD in individuals with and without CKD at baseline are still lacking [113].

### 6.2. Studies in Pediatric Populations

The estimated prevalence of MAFLD in overweight or obese children is 33.78% [114]. As in adults, it is associated with atherosclerosis, dyslipidemia, insulin resistance, and hypertension [25,115,116]. Despite the similarities to adult MAFLD with regard to epidemiology and pathogenesis, some histological differences are found [92]. For instance, lipid accumulation in children occurs preferentially in zone 1 hepatocytes or in a panacinar fashion and is usually associated with portal fibrosis and inflammation rather than ballooned hepatocytes and perisinusoidal fibrosis, as seen in adults [92,116]. Progression to steatohepatitis and cirrhosis may occur, but the incidence of these conditions remains under-investigated [117,118]. Considering these differences and due to the scarcity of studies involving pediatric populations, the risk of children and adolescents developing CKD is largely unknown [25].

Pacifico et al. compared the prevalence of low eGFR or abnormal albuminuria among 268 overweight/obese children with NAFLD, 328 overweight/obese children without NAFLD, and 130 healthy normal-weight controls. NAFLD was independently associated with impaired eGFR and/or microalbuminuria (OR = 2.54 [95% CI: 1.16, 5.57]) [119]. In contrast, another study compared kidney function in 80 children with biopsy-proven NAFLD and 59 children of normal weight matched for age and sex. Despite higher levels of insulin resistance in the NAFLD group, albuminuria and creatinine clearance did not differ significantly between groups [84].

In terms of NAFLD severity, a recent study involving 1164 children with biopsy-proven NAFLD/MASLD found that the prevalence of glomerular hyperfiltration was independently associated with significant liver fibrosis (OR = 1.45), but the incidence of kidney function deterioration did not differ according to NAFLD/MASLD severity in 2 years of follow-up [74].

Therefore, more prospective studies involving pediatric patients and longer follow-up periods are needed, preferentially exploring biomarkers of MASLD specific for this age group [120].

### 6.3. Impact of Alcohol Intake and MetALD

As alcohol intake and metabolic syndrome are common throughout the world, steatotic liver disease of a dual etiology (now termed MetALD) is expected to be frequent [121,122]. Studies have shown that MAFLD combined with alcohol consumption increases the risk of insulin resistance, T2DM, liver injury, and hepatic fibrosis in a dose-dependent manner according to the number of drinks per week and number of cardiometabolic factors [122,123,124]. From a mechanistic viewpoint, therefore, the risk of CKD would be expected to be higher in such individuals. Nevertheless, the impact of MetALD on kidney function remains unclear [26].

One study showed that moderate alcohol intake was in fact a protective factor for CKD in individuals with NAFLD (OR = 0.37 [95% CI 0.22, 0.65]), even after adjusting for confounding factors [125]. Along the same line, a meta-analysis of 15 cohort studies showed that individuals with moderate and high alcohol intake had a 24% (RR = 0.76 [95% CI: 0.70, 0.83]) and 21% (RR = 0.79 [95% CI: 0.71, 0.88]) lower risk of CKD, respectively, compared to non-drinkers or occasional drinkers [126]. Conversely, another study reported a higher risk of incident CKD in individuals with MASLD (HR = 1.20 [95% CI: 1.08, 1.33]) and those with ALD (HR = 1.41 [95% CI: 1.05, 1.88]) but not those with MetALD (HR = 1.11 [95% CI: 0.90, 1.36], *p* = 0.332) after adjusting for age, sex, eGFR, current smoking habit, T2DM, systemic hypertension, and dyslipidemia [78].

Considering these conflicting findings, further studies are needed to clarify the association between MetALD and CKD.

## 7. Treatment Options for Both MASLD and CKD

The management of MASLD and CKD is mainly focused on weight loss and lifestyle changes through exercise and hypocaloric diets [127]. These measures have been associated with improvements in liver inflammation and fibrosis [127,128].

In severe obesity, bariatric surgery has also proved to be associated with the regression of steatohepatitis, an improvement in the glomerular filtration rate, and lower levels of albuminuria [129]. Exercise is also recommended to avoid sarcopenia [28].

Resmetirom, which is a thyroid hormone receptor-beta (THR-beta) agonist, is the first drug approved by the US Food and Drug Administration for the treatment of NASH. In a phase 3 randomized controlled trial, resmetirom showed greater efficacy compared to placebo in terms of resolution of NASH and improvement in liver fibrosis [130]. The impact of this drug on kidney function remains to be investigated [131].

Less specific pharmacotherapy has also been used in the management of metabolic dysfunction-related diseases and cardiovascular diseases, with varying degrees of beneficial effects on both CKD and MAFLD [19]. Some classes of drugs that have shown favorable results in this context are renin–angiotensin system inhibitors, PPAR-γ agonists, incretin receptor agonists, and glucagon-like peptide receptor agonists [19,132].

New drugs targeting other pathogenic mechanisms are under investigation. Farnesoid X receptor agonists have been associated with a regression in both kidney and liver diseases by reducing inflammation, oxidative stress, fibrosis, and apoptosis [46]. Gut microbiome modulators, such as prebiotics and TMAO inhibitors, have shown conflicting results in preclinical studies, and better definitions as to their efficacy are awaited [20]. Randomized control trials involving patients treated for MASLD and investigating the development of CKD as an endpoint are also warranted [7].

## 8. Conclusions and Future Directions for Research

MASLD and CKD have some risk factors in common, but MASLD may directly induce CKD. This association has been elucidated, but limitations in the literature still need to be overcome, especially the gaps in research involving specific groups of patients. The recently adopted nomenclature—which is based on clearly defined subgroups—is likely to make inclusion criteria more homogeneous in future research. Given the systemic repercussions of MASLD, it is crucial that clinicians remain vigilant in monitoring patients with MASLD as a potential high-risk group for CKD. Early identification and intervention could be key in preventing the progression of both diseases.

Future research should focus on several important areas. First, a deeper understanding of the pathogenesis of MASLD on a cellular and molecular level is needed to identify the precise mechanisms by which it may lead to kidney damage. This will allow for the development of targeted treatments that can mitigate its harmful effects on the kidneys.

Recent diagnostic advances in MASLD are likely to enhance the precision of inclusion and exclusion criteria in future research. Among non-invasive tests, MRI and serum biomarkers are likely to become both more affordable and accurate as technologies continue to evolve [102,133]. Cutting-edge approaches include AI-assisted histological analysis, which enhances diagnostic consistency in biopsy interpretation, and multi-omic profiling (e.g., metabolomics, proteomics) for identifying molecular signatures of disease progression [134].

An important area requiring attention is the pediatric population. Although MASLD is increasingly recognized in children and adolescents, its long-term renal consequences remain poorly defined. Future studies should explore early biomarkers of kidney injury in pediatric MASLD, to prevent the transition from early metabolic liver disease to multi-organ dysfunction in adulthood [25].

Another emerging priority is MetALD, a recently defined entity that combines features of metabolic dysfunction with alcohol-induced liver injury. The interplay between MetALD and kidney disease is largely unexplored but may be clinically significant, particularly as both alcohol use and metabolic disorders frequently coexist. Future research should investigate whether MetALD confers an additive or synergistic risk for CKD compared to MASLD or alcohol-related liver disease alone [135].

Lastly, as the global prevalence of obesity continues to rise, and as populations age, the burden of both MASLD and CKD is expected to increase. Addressing these growing public health challenges will require a concerted effort, involving not only scientific research but also public health strategies focused on promoting healthy lifestyles and mitigating the effects of an increasingly sedentary and nutritionally poor environment [28,136]. Clinicians and researchers alike must work together to clarify the full scope of the interplay between MASLD and CKD and to develop effective therapeutic strategies to combat this dual disease burden.

## Figures and Tables

**Figure 1 biomedicines-13-02162-f001:**
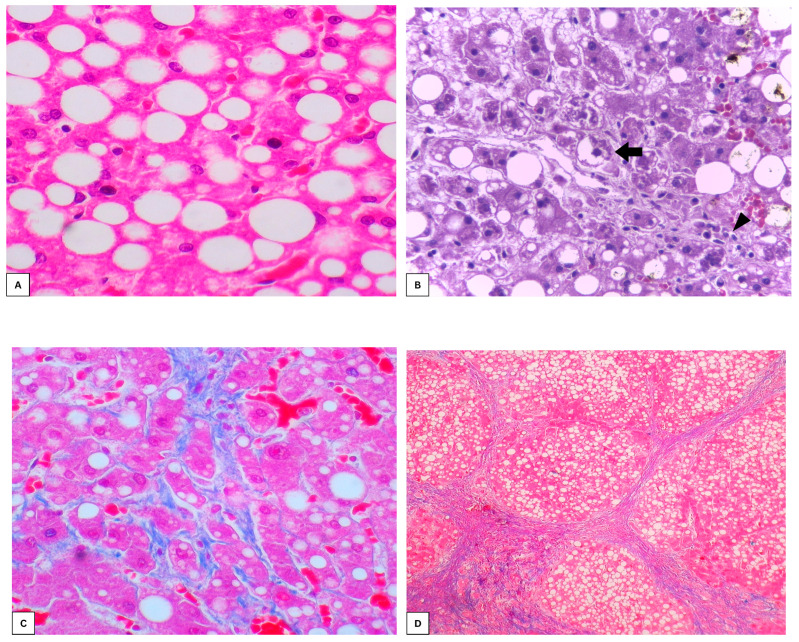
Histological progression of MASLD. (**A**): Isolated hepatic steatosis (Hematoxylin/eosin (H.E), 400× magnification); (**B**): steatohepatitis, with ballooned hepatocytes in the center (black arrow) and inflammatory infiltrate (arrowhead) (H.E, 400×
magnification); (**C**): early perisinusoidal fibrosis (blue dye) around hepatocytes (Masson’s trichrome, 400× magnification); (**D**): cirrhosis, with fibrous septa (blue dye) and steatotic regenerative nodules, surrounded by these septa (Masson’s trichrome, 40× magnification). Source: original photomicrograph from the pathology archive, Federal University of Pernambuco (*UFPE*).

**Figure 2 biomedicines-13-02162-f002:**
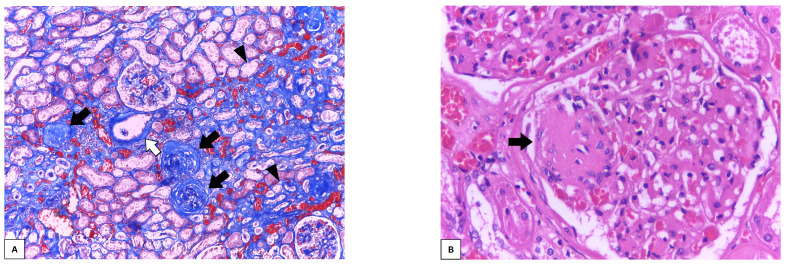
Histology of CKD. (**A**): (black arrows) globally sclerotic glomeruli; (arrowheads) interstitial fibrosis surrounding atrophic tubules; (white arrow) arterial intimal thickening (Masson’s trichrome, 100x magnification). (**B**): (black arrow) nodular glomerulosclerosis, characteristic of diabetic nephropathy (H.E., 400× magnification). Source: original photomicrograph from the pathology archive, Federal University of Pernambuco (*UFPE*).

**Figure 3 biomedicines-13-02162-f003:**
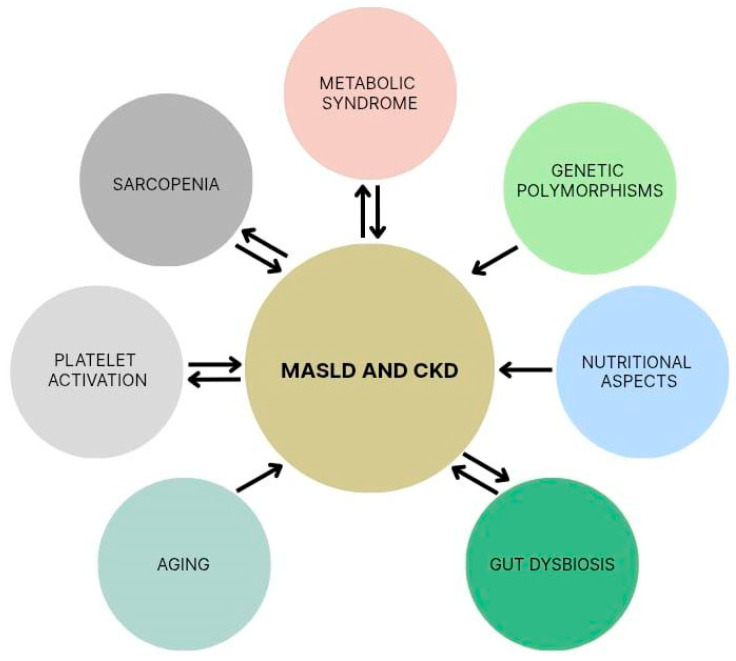
Shared risk factors for metabolic dysfunction-associated steatotic liver disease (MASLD) and chronic kidney disease (CKD). Double arrows indicate a bidirectional relationship.

**Figure 4 biomedicines-13-02162-f004:**
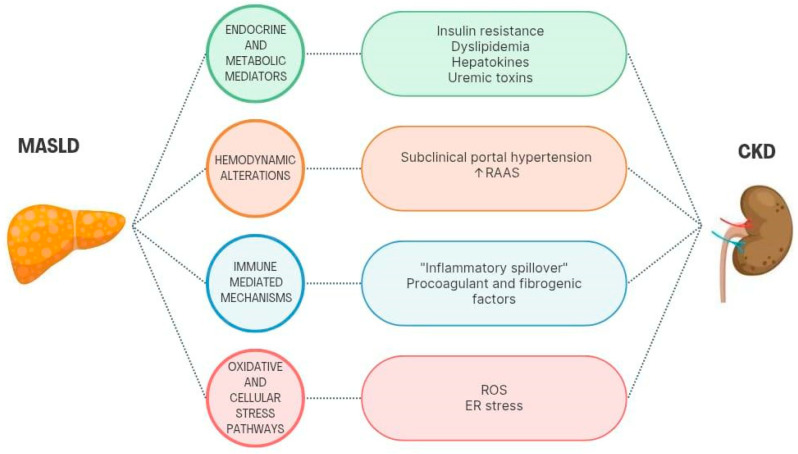
Hypothetical mechanisms of direct MASLD-induced CKD. The main groups of hepatic-induced disturbances are represented in the circles. Each of them is exemplified by the dysregulations displayed in the rounded rectangles, all contributing to the pathogenesis of CKD.

**Table 1 biomedicines-13-02162-t001:** Impacts of shared risk factors on MASLD and CKD, along with bidirectional effects (impact of MASLD and CKD on each risk factor).

Risk Factor	Effect on MASLD/Pathogenic Mechanisms	Effect on CKD/Pathogenic Mechanisms	Bidirectional Effect/Pathogenic Mechanisms
**Metabolic Syndrome**	Increases steatosis, MASH, and fibrosis, through ectopic fat accumulation, inflammation, oxidative stress, fibrogenic factors, dysregulation of PPAR-γ, ↑ leptin, ↓ adiponectin, and increased insulin resistance [45,46,47,48,49].	Direct parenchymal compression by visceral fat, renal artery stenosis, glomerulosclerosis, proteinuria, ↓ GFR, and microvascular disease through ectopic fat accumulation, atherogenic dyslipidemias, activation of the RAAS, and ↑ advanced glycation end-products [11,21,50].	MASLD and CKD worsen metabolic syndrome by increasing insulin resistance, dyslipidemia, and hypertension through inflammation, oxidative stress, uremia, metabolic acidosis, sedentarism, and further activation of the RAAS [7,47,51].
**Genetic Polymorphisms**	Mutations in PNPLA3, TM6SF2, MBOAT7, and GCKR genes cause steatosis/steatohepatitis and liver fibrosis through fat deposition and inflammation [52,53].	These mutations promote glomerular and tubular lesions through lipotoxicity, inflammation, and insulin resistance [52,53].	Not applicable
**Nutritional aspects**	Hypercaloric and fructose-rich diets promote steatosis through insulin resistance and dyslipidemia [20,44].	This diet induces nephrotoxicity through hyperuricemia, insulin resistance, and dyslipidemia [7,46].	Not applicable
**Gut Dysbiosis**	Toxins (LPS, TMAO) produced by harmful bacteria reach portal circulation and stimulate liver inflammation and fibrosis through activation of TLR, Kupffer cells, and stellate cells [8,54,55].	LPS and TMAO trigger renal atherosclerosis and interstitial fibrosis through activation of TLR, oxidative stress, and inflammatory pathways [20,21].	Hepatotoxins and increased urea levels in systemic circulation promote further intestinal absorption of LPS and TMAO through disruption of the gut barrier and increased permeability [54,56].
**Aging**	Associated with steatosis/steatohepatitis and fibrosis through mitochondrial dysfunction, low antioxidant capacity, and damage to sinusoidal cells [41,42].	Renal artery stenosis and interstitial fibrosis through calcification, ↑ fetuin-A, ↓ adiponectin, and ↑ oxidative stress [38,57].	Not applicable
**Platelet Activation**	Promotes liver fibrosis and inflammation through cytokines (IL-6, TNF-alpha) and growth factors (PDGF, IGF-1, FGF, and TGF-beta) [58,59,60].	Kidney fibrosis and atherosclerosis through these cytokines and growth factors [59,60].	Parenchymal damage in both organs activates platelets through endothelial activation, oxidative stress, and systemic liberation of uremic toxins [13,40,60].
**Sarcopenia**	Induces liver inflammation and fibrosis through activation of hepatic stellate cells caused by myostatin liberation by sarcopenic muscles [61].	Worsens renal function through systemic inflammation and insulin resistance [19].	Reduced muscle mass through liver catabolism of amino acids and production of cytokines and ammonia (in case of cirrhosis). Additional sarcopenia through CKD-induced systemic inflammation, protein loss in dialysis, protein catabolism, and reduced anabolism, along with metabolic acidosis and sedentarism [40,62,63,64].

Abbreviations: MASH, metabolic dysfunction-associated steatohepatitis; PPAR-gama, peroxisome proliferator-activated receptor gammaγ; GFR, glomerular filtration rate; MASLD, metabolic dysfunction-associated steatotic liver disease; CKD, chronic kidney disease; RAAS, renin–angiotensin–aldosterone system; PNPLA3, patatin-like phospholipase domain-containing protein 3; TM6SF2, Transmembrane 6 superfamily member 2; MBOAT7, Membrane-bound O-acyltransferase domain-containing 7; GCKR, clucokinase regulator; LPS, lipopolysaccharide; TMAO, trimethylamine N-oxide; TLR, Toll-like receptor; IL-6, interleukin-6; TNF-alpha, tumor necrosis factor-alpha; PDGF, platelet-derived growth factor; IGF-1, insulin-like growth factor 1; FGF, fibroblast growth factor; TGF-beta, transforming growth factor-beta.

**Table 2 biomedicines-13-02162-t002:** Main meta-analyses and longitudinal studies on the association between MASLD and CKD.

Authors, Year	Study Characteristics	Diagnosis of SLD/CKD	Main Findings	Comments
Liu et al., 2024 [73]	Systematic review and meta-analysis of 8 cohort studies from Asia and the UK, comprising 598,531 patients. Follow-up ranging from 4.6 to 12.9 years. Aim: to compare the incidence of CKD in persons with MAFLD and controls without MAFLD.	SLD: MAFLD criteria, diagnosed by ultrasound. CKD: eGFR < 60 mL/min/1.73 m^2^, proteinuria, or urine albumin/creatinine ratio ≥ 30 mg/g.	MAFLD was associated with a higher incidence of CKD (HR: 1.38, 95% CI: 1.24–1.53) adjusted for sex, body mass index, cardiovascular disease, diabetes, hypertension, and smoking status.	1. Meta-analysis with high heterogeneity (I^2^ = 95%). 2. Seven of the eight studies were from Asia. 3. No assessment of MAFLD severity at baseline.
Agustanti et al., 2023 [14]	Systematic review and meta-analysis of 11 studies (7 cross-sectional and 4 longitudinal) from Asia, Europe, and the USA, comprising 355,886 patients. Follow-up ranging from 4.6 to 6.5 years. Aim: to determine the incidence and prevalence of CKD according to the presence and severity of MAFLD at baseline.	SLD: MAFLD or NAFLD criteria, diagnosed by ultrasound, transient elastography, or FLI. CKD: eGFR < 60 mL/min/1.73 m^2^ or UACR of 30 mg/g or greater or proteinuria (positive dipstick test result of +1 or greater).	MAFLD was associated with higher prevalence (OR = 1.50, 95% CI 1.02–2.23) and incidence (HR = 1.35, 95% CI 1.18–1.52) of CKD adjusted for age, sex, comorbidities, study region, and follow-up duration. Significant liver fibrosis, but not steatosis, was associated with greater likelihood of developing CKD.	1. Meta-analysis with high heterogeneity (I^2^: 97.7% for cross-sectional and 84.6% for longitudinal studies). 2. The exposure group included patients with either NAFLD or MAFLD criteria, leading to possible selection bias. 3. Absence of histological analysis of SLD.
* Mantovani et al., 2022 [16]	Systematic review and meta-analysis of 13 longitudinal studies from Asia, Europe, and the USA, comprising 1,222,032 patients. Median follow-up of 9.7 years. Aim: to determine the incidence of CKD according to the presence and severity of MAFLD at baseline.	SLD: NAFLD criteria, diagnosed by liver enzymes, blood biomarkers, imaging methods, liver histology, or ICD-10 codes. CKD: eGFR < 60 mL/min/1.73 m^2^ with or without overt proteinuria.	NAFLD was associated with higher incidence of CKD (HR = 1.43, 95% CI 1.33–1.54) adjusted for age, sex, obesity, hypertension, diabetes, and other conventional CKD risk factors.	The authors suggest a possible association between the severity of NAFLD (especially liver fibrosis) and incident CKD but emphasize that the studies that assessed hepatic fibrosis did not include a control group without NAFLD, resulting in insufficient data for a meta-analysis.
* Musso et al., 2014 [15]	Systematic review and meta-analysis of 33 studies (20 cross-sectional and 13 longitudinal) from Asia, the USA, Europe, and Saudi Arabia, comprising 63,902 patients. Follow-up ranging from 3 to 27 years. Aim: to determine the incidence and prevalence of CKD according to the presence and severity of NAFLD at baseline.	SLD: NAFLD criteria, diagnosed by liver histology, imaging (ultrasound, computer tomography, magnetic resonance imaging, spectroscopy), or biochemistry (elevations in serum liver enzymes). CKD: eGFR < 60 mL/min/1.73 m^2^, proteinuria (UACR, 24 h albumin excretion rate, fresh morning urine dipstick), or other abnormalities due to tubular disorders or structural abnormalities detected by electrolyte or urinary sediment alterations, histology, imaging, or history of kidney transplantation.	NAFLD was associated with higher prevalence (OR = 2.12, 95% CI 1.69–2.66) and incidence (HR = 1.79, 95% CI 1.65–1.95) of CKD. NASH was associated with higher prevalence (OR = 2.53, 95% CI 1.58–4.05) and incidence (HR = 2.12, 95% CI 1.42–3.17) of CKD than steatosis alone. Advanced liver fibrosis was associated with higher prevalence (OR = 5.20, 95% CI 3.14–8.61) and incidence (HR = 3.29, 95% CI 2.30–4.71) of CKD than non-advanced fibrosis. All findings were adjusted for diabetes status, traditional risk factors for CKD, obesity, and insulin resistance.	Only 5 studies with biopsy-proven NAFLD, totaling 690 patients, leading to possible small study bias.
Sanyal, 2021 [76]	Prospective multicenter cohort study from the USA, including 1773 patients with NAFLD (with or without fibrosis). Median follow-up of 4 years. Aim: to determine longitudinal outcomes according to the severity of NAFLD.	SDL: NAFLD criteria, diagnosed by liver biopsy. CKD: decrease in eGFR of >40%.	Patients with stage F4 fibrosis had a decrease of more than 40% in the eGFR compared to those with stages F0 to F2 fibrosis (2.98 vs. 0.97 events per 100 person-years; HR = 1.9; 95% CI 1.1–3.4) adjusted for age, race, sex, length of biopsy specimen, and diabetes status.	1. Limited generalizability (study conducted at tertiary care centers with predominantly White populations. 2. F3 fibrosis was not associated with a decrease in eGFR of >40% when compared to F0-F2 (HR = 0.9; 95% CI 0.6–1.6).
Gao et al., 2024 [80]	Retrospective cohort study from China, including 79,540 patients. Median follow-up of 12.9 years. Aim: to determine the incidence of CKD according to the presence, severity, and remission of MAFLD/MASLD.	SLD: MAFLD and MASLD criteria, diagnosed by ultrasound. CKD: eGFR < 60 mL/min/1.73 m^2^ or positive proteinuria (≥1+).	MAFLD/MASLD was associated with a higher incidence of CKD (HR, 1.12 [95% CI, 1.09–1.16]); risk increased according to severity of steatosis (*p* < 0.001). Even after remission of MAFLD/MASLD, patients with prior moderate to severe hepatic steatosis still had a higher risk of CKD. Adjustments: age, sex, smoking, drinking, exercise, education, income, eGFR at baseline, uric acid, ALT, metabolic dysfunction, use of antihyperglycemic, antihypertensive, and antilipidemic agents.	Most patients in the study were male (80%).
Mori et al., 2024 [78]	Retrospective cohort study from Japan, including 12,138 patients. Follow-up of 10 years. Aim: to determine the incidence of CKD according to SLD status at baseline (MASLD, MetALD, ALD, and SLD without metabolic dysfunction) compared to the incidence of CKD in non-SLD control subjects.	SLD: MASLD, MetALD, and ALD criteria, diagnosed by ultrasound. CKD: eGFR < 60 mL/min/1.73 m^2^ or proteinuria by dipstick.	The incidence of CKD was higher in individuals with MASLD (HR = 1.20; CI 1.08–1.33) and ALD (HR = 1.41; CI 1.05–1.88), but not MetALD (HR = 1.11; CI 0.90–1.36), when compared to those without SLD. Individuals with SLD without metabolic dysfunction had a lower incidence (HR = 0.61 [0.39–0.96]) than those without SLD. Adjustments: age, sex, baseline eGFR, smoking, diabetes, systemic hypertension, and dyslipidemia.	Study conducted in a single urban clinic (possibility of selection bias).
* Liang et al., 2022 [77]	Prospective cohort study from China, including 6873 patients. Follow-up of 4.6 years. Aim: to compare the incidence of extra-hepatic diseases in persons with MAFLD and controls without MAFLD.	SLD: NAFLD and MAFLD criteria, diagnosed by ultrasound. CKD: eGFR < 60 mL/min/1.73 m^2^ or UACR 30 μg/mg or greater.	MAFLD was associated with a higher incidence of CKD (RR = 1.64; 95% CI 1.39–1.94). Similar associations for NAFLD were observed. Adjustments: age, sex, education, smoking status, and leisure-time exercise.	NAFLD and MAFLD were also associated with an increased incidence of diabetes and cardiovascular diseases.
* Mouzaki et al., 2024 [74]	Prospective multicenter cohort study from the USA, including 1164 children (age 13 ± 3 years). Median follow-up of 2 years. Aims: to determine the prevalence of hyperfiltration or CKD in children with NAFLD/MASLD and determine links with liver disease severity.	SLD: MASLD/NAFLD criteria, diagnosed by biopsy. CKD: hyperfiltration or eGFR < 90 mL/min/1.73 m^2^.	Significant liver fibrosis was associated with hyperfiltration (OR: 1.45). Progression of renal impairment was not associated with change in liver disease severity. Adjustments: BMI, insulin resistance, hemoglobin A1c, blood pressure, age, ethnicity, race, gender, and T2DM.	1. Most participants were Hispanic. 2. Short follow-up. 3. Study did not include proteinuria as a definition of CKD.
* Zuo et al., 2021 [75]	Prospective cohort study from China, including 4042 patients aged 40 years or more. Mean follow-up of 4.4 years. Participants were divided into 4 groups at baseline: 57.4% participants with non-NAFLD, 13.2% with incident NAFLD, 21.6% with persistent NAFLD, and 7.8% with NAFLD resolution. Aim: to assess associations between changes in NAFLD status/progression of NAFLD fibrosis and the risk of incident CKD.	SLD: NAFLD criteria, diagnosed by ultrasound and NAFLD fibrosis score. CKD: UACR 30 mg/g or greater, or eGFR 60 mL/min/1.73 m^2^ or lower.	Incident NAFLD was associated with incident CKD (OR = 1.44; 95% CI, 1.003–2.06) compared to non-NAFLD. However, the risk of incident CKD was not significantly different between groups with NAFLD resolution and persistent NAFLD. In the persistent NAFLD group, fibrosis progression was associated with a higher incidence of CKD compared to stable fibrosis (OR = 2.82; 95% CI, 1.22–6.56). Adjustments: baseline and evolution of diabetes, hypertension, and obesity.	1. The study population consisted of more women and older participants from a community-based population. 2. NAFLD and CKD at baseline and follow-up were determined concurrently, precluding the determination of causation.
* Heo et al., 2024 [79]	Prospective cohort study from Korea, including 214,145 adults with normal kidney function at baseline. Median follow-up: 6.1 years. Aim: to compare the incidence of CKD among 5 groups of participants: without steatosis, NAFLD-only, MASLD-only, both NAFLD and MASLD, and SLD not categorized as NAFLD or MASLD.	SLD: NAFLD and MASLD criteria, diagnosed by ultrasound. CKD: eGFR < 60 mL/min/1.73 m^2^ or albuminuria.	The group meeting both NAFLD and MASLD criteria had the highest risk of CKD (HR = 1.21 [95% CI, 1.04–1.42]). The MASLD-only group had a similar risk (HR = 1.96 [95% CI, 1.44–2.67]), but NAFLD alone was not independently associated with CKD or albuminuria. Adjustments: age, sex, education, smoking history, exercise, alcohol intake, history of coronary artery disease, use of anti-hypertensive medications, and levels of eGFR at baseline.	Authors suggest that MASLD criteria are better than NAFLD criteria at identifying individuals at high risk of incident CKD or albuminuria.

Abbreviations: SLD, steatotic liver disease; FLI, fatty liver index; CKD, chronic kidney disease; CI, confidence interval; eGFR, estimated glomerular filtration rate; MASLD, metabolic dysfunction-associated steatotic liver disease; HR, hazard ratio; MAFLD, metabolic dysfunction-associated fatty liver disease; NAFLD, non-alcoholic fatty liver disease; ICD-10, International Classification of Diseases—tenth revision; UACR, urinary albumin-creatinine ratio; T2DM, type 2 diabetes mellitus; OR, odds ratio; RR, relative risk; ALD, alcoholic liver disease. * denotes meta-analyses with lower heterogeneity and greater generalizability, along with the most robust prospective studies.

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
