# Peer review of "Metabolic Dysfunction-Associated Steatotic Liver Disease as a Risk Factor for Chronic Kidney Disease: A Narrative Review"

_biomedicines, 2025, doi:10.3390/biomedicines13092162_

Round 1
Reviewer 1 Report
Comments and Suggestions for Authors
The manuscript is timely, comprehensive, and well-organized. It effectively outlines the evolving terminology, pathophysiological mechanisms, and clinical associations between MASLD and CKD. However, minor structural and clarity improvements are recommended.
In Introduction part, the rationale for replacing NAFLD/MAFLD with MASLD is well laid out. You could consider briefly explaining why MASLD is a more precise term even before detailed nomenclature discussion.
In Search Strategy (Section 2), Clarify whether any inclusion/exclusion criteria (e.g., human studies only, age range, language) were applied to screen papers beyond the keywords.
In Shared Risk Factors (Section 4), include a summary table or flow diagram summarizing all shared risk factors and their bidirectional effects on MASLD and CKD to enhance clarity. Some references used are over a decade old; ensure all cited works reflect the latest available literature where possible.
In Mechanistic Pathways (Section 5), the figures and mechanistic explanations (e.g., endocrine, hemodynamic, inflammatory pathways) are informative. Consider integrating Figure 4 earlier to provide a visual reference before diving deep into individual mechanisms.
In Table 1, include a brief concluding paragraph summarizing patterns seen across studies (e.g., severity of MASLD → increased CKD risk). Consider color-coding or symbols to denote high-quality studies vs. those with high heterogeneity or limited generalizability.
In Gaps and Limitations (Section 6), reduce repetition; some limitations (e.g., diagnostic heterogeneity) are mentioned in both subsections 6.1 and 6.3. Include brief proposals or tools that could address limitations (e.g., prospective cohort designs, multi-omics, pediatric biomarkers).
Figures are informative but would benefit from higher resolution and consistent labeling (e.g., colors or arrows should be described in legends).
Reviewer 2 Report
Comments and Suggestions for Authors
The review is well written, covering various mechanisms that could lead from MASLD to CKD. Despite that, certain observations impose.
By conceptualizing the manuscript under the title "Metabolic Dysfunction-Associated Steatotic Liver Disease as a Risk Factor for Chronic Kidney Disease: A Narrative Review," the authors aim to provide a comprehensive overview of the existing literature on the association between MASLD and CKD. MASLD is characterized by abnormal fat accumulation in the liver, representing a systemic disorder of lipid metabolism. Given the liver's central role in regulating metabolic processes, hepatic dysfunction caused by pathological fat deposition may adversely affect the function of other organs, including the kidneys. The topic of the review is relevant, since the obesity epidemic is on rise, and MASLD and CKD is more prevalent in these individuals.
This review presents current findings on shared risk factors between MASLD and CKD, while also exploring the potential of MASLD to act as a direct etiological factor in the development of CKD. Notably, the authors critically address limitations within the existing literature, such as variability in study design, terminology, inclusion criteria, limited data on pediatric populations, and the confounding effects of alcohol consumption.
Overall, the manuscript is well-structured and informative. With minor revisions to address the noted inconsistencies, it is a strong candidate for publication.
The authors did not cite the sources of the first two figures (histological preparations).
I do not understand why section 6.4 falls under Gaps and Limitations in Literature.
In row 54, the letter f in figure 1 should be capitalized.
